# Sodium Benzoate and Potassium Sorbate Inhibit Proteolysis and Promote Lipid Oxidation in Atlantic Herring Marinades Produced on an Industrial Scale

**DOI:** 10.3390/molecules30204103

**Published:** 2025-10-15

**Authors:** Mariusz Szymczak, Patryk Kamiński, Barbara Szymczak, Ingrid Undeland, Izabela Dmytrów

**Affiliations:** 1Department of Toxicology, Dairy Technology and Food Storage, Faculty of Food Sciences and Fisheries, West Pomeranian University of Technology in Szczecin, Pawła VI Street No. 3, 71-459 Szczecin, Poland; patryk-kaminski@zut.edu.pl (P.K.); izabela.dmytrow@zut.edu.pl (I.D.); 2Department of Applied Microbiology and Human Nutrition Physiology, Faculty of Food Sciences and Fisheries, West Pomeranian University of Technology in Szczecin, Pawła VI Street No. 3, 71-459 Szczecin, Poland; barbara.szymczak@zut.edu.pl; 3Department of Life Sciences–Food and Nutrition Science, Chalmers University of Technology, SE 412 96 Gothenburg, Sweden; undeland@chalmers.se

**Keywords:** marinated herring, sodium benzoate, potassium sorbate, proteases, lipid oxidation, microbiology

## Abstract

Cold-ripened fish marinades, produced mainly from Atlantic herring, represent one of the major seafood products in Northern and Central Europe. Because the shelf-life of these mildly acidified, salty products rarely surpasses 4 weeks, more than half of the commercial lots contain the preservatives sodium benzoate (E211) and potassium sorbate (E202). However, the broader technological consequences of such additives remain poorly documented. This study evaluated the impact of 0.25 (*w*/*w*) benzoate + 0.10 g/100 g sorbate on the quality of industrial-scale marinades (200 kg fish; 7 d, 4 ± 1 °C). Physicochemical traits (mass loss, pH, proximate composition, salt content, colour, and texture), enzymatic indices of ripening (cathepsins, amino-peptidases, and TCA soluble nitrogen fractions), lipid oxidation, microbial growth, and sensory attributes were analyzed. Preservatives caused only marginal changes in pH and proximate composition (0.3–3.4% *w*/*w* differences) but markedly suppressed proteolysis. Free amino acid and peptide fractions in muscle decreased by 6.0% and 8.8%, in parallel to 45% and 22% reductions in leucine- and alanine-amino-peptidase activities in muscle. In the marinating brine, the levels of total nitrogen, peptides, and free amino acids were also lower in the samples with preservatives, confirming that sodium benzoate and potassium sorbate slowed down the enzymatic ripening of the marinades. Concomitantly, peroxide, p-anisidine, and TOTOXs increased by up to 9.4, 71.3, and 33.7%, respectively, indicating accelerated lipid oxidation despite the chelating capacity of benzoate/sorbate acids. Overall sensory acceptability declined slightly (−0.15 points on a five-point scale), mainly owing to chemical off-flavours and lower juiciness. Microbial counts remained <1.0 log CFU/g in the preservative batch versus 2.1 log in the control. Benzoate–sorbate combinations effectively stabilized the microbiota of marinated herring without appreciably altering basic physicochemical traits, but they retard enzymatic ripening, diminish antioxidant peptide pools, and thereby promote lipid oxidation—collectively lowering the nutritional value. The data supports a cautious, minimal-use approach to application of chemical preservatives in cold-ripened fish products.

## 1. Introduction

Herring is among the most commonly processed fish species worldwide, with marination being a traditional method for preservation and flavour enhancement. On the market, three main types of marinades are available: cold-ripened, boiled, and fried, with cold marinades accounting for over 95% of total production. Poland is one of the leading European exporters of marinated herring, producing approximately 100,000 tons annually. A significant proportion of these products, especially those exported to the United States and distributed within Europe, contain chemical preservatives. Currently, fish processors in the European Union have announced an increased use of preservatives for marinated and salted herring due to the introduction of zero-tolerance regulations for *L. monocytogenes* in ready-to-eat food [1]. Their inhibitory effect is based on the penetration of undissociated acid molecules into microbial cells, causing intracellular acidification, enzyme inhibition and disruption of metabolic pathways, which is particularly effective against *L. monocytogenes* and thereby extends the shelf life of marinated herring.

The marination process involves soaking herring fillets in a mixture of water, acetic acid, and salt at temperatures between 0 and 10 °C. The ripening phase may last from several days to several months. The diffusion of acetic acid and salt into the muscle tissue results in a decrease in pH and an increase in osmotic pressure [2]. This environment reduces microbial growth at the same time as it activates endogenous proteolytic enzymes (cathepsins) that hydrolyze proteins into peptides and free amino acids contributing to the characteristic flavour, aroma, and texture of marinated herring. The product becomes ready for consumption without requiring further thermal processing.

Although mildly marinated herring typically contains 1.0–2.5 g/100 g acetic acid and 2.0–3.5 g/100 g salt, these concentrations are insufficient to ensure full microbial stability during extended refrigerated storage [3]. Increasing the levels of acid and/or salt can extend shelf life but often results in reduced sensory acceptability [3]. The addition of saccharose up to 15 g/100 g in brine and organic acids other than acetic acid has also been shown to shorten shelf life [4]. To address the challenges posed by seasonal fluctuations in supply and demand, marinated herring is often produced months in advance of its sales, necessitating extended storage times. For this reason, chemical preservatives such as sodium benzoate (E 211) and potassium sorbate (E 202) are used in approximately 60% of the marinades, particularly those packed with sauces, vegetable oil, and malic or tartaric acid as cover-brine instead of vinegar.

Sodium benzoate and potassium sorbate are low-cost, water-soluble preservatives with broad-spectrum antimicrobial activity, especially in the pH range of 2.5–4.5, within which they remain predominantly in their undissociated active form [5]. In marinades, both preservatives are often used together because they complement each other in terms of their antimicrobial spectrum. Potassium sorbate is more effective against mould, Gram-positive bacteria, and lactic acid bacteria, while sodium benzoate exhibits stronger activity against yeasts and Gram-negative bacteria. Both preservatives are highly effective against *L. monocytogenes* at pH values below 4.5, salt concentrations of 2–3 g/100 g, and under refrigerated conditions [6,7]. They readily diffuse into the fatty muscle tissue of herring and do not affect the product’s appearance. Their use is widespread not only in marinated fish but also in salted, smoked, vacuum-packed, and modified-atmosphere packaged seafood [8]. However, the majority of previous studies comprising the role of these preservatives for the shelf life of seafood have not taken into account the unique physicochemical conditions of fish marinades and their interaction with endogenous muscle proteases responsible for ripening.

Despite their long-standing use in the marination industry, the impact of sodium benzoate and potassium sorbate on the proteolytic activity and sensory attributes of marinated herring remains poorly understood. Most of the available literature on this topic dates back to the 1960s–1980s and has revealed that preservatives have effects on microbial quality, sensory evaluation, protein content, and overall proteolysis [4,9,10]. In recent years, seafood preservation research has increasingly focused on reducing chemical additives and exploring natural or process-based stabilization methods. Nevertheless, there is still limited understanding of how conventional preservatives influence endogenous biochemical processes in fish muscle, particularly protease activity and lipid oxidation, which are crucial for texture and nutritional quality. The present study addresses this gap by examining the relationship between the preservative-induced inhibition of proteolysis and lipid oxidation in industrially produced marinated herring. To our knowledge, previous studies have not described any effect of preservatives on herring lipids and muscle proteases. Therefore, we hypothesized that preservatives can be good inhibitors of muscle proteases, which in turn could affect the lipid quality. Therefore, this study aimed to evaluate the effects of sodium benzoate and potassium sorbate, within levels allowed by EU Regulation (EC) No 1333/2008, and consistent with the scientific opinion of European Food Safety Authority (EFSA, 2015) [11], on the physicochemical properties, proximate composition, enzymatic activity, microbial stability, lipid oxidation development, and sensory attributes of marinated herring fillets.

## 2. Results and Discussion

Herring marinade manufacturers employ different concentrations of sodium benzoate and potassium sorbate, very often approaching the maximum permitted by EU Regulations and EFSA law. The advantage of these preservatives, especially for low pH foods, is that they diffuse rapidly from the brine into the herring muscle. The diffusion equilibrium between the brine and the muscle is reached after 3 days, but the exact time depends on the pH as well as the water and fat content of the herring muscle [10]. After 7 days of ripening, the preservative content in the herring muscle was found to be twice as high as that in the marinating brine [10]. This suggests that the concentrations of sodium benzoate and potassium sorbate in 100 g of the herring muscle tested in the present study were at least 125 mg and 50 mg, respectively, corresponding to about 1250 ppm and 500 ppm, which is within the range typically found in marinated fish products [10].

### 2.1. Physical Properties and Proximate Composition

Marinated fillets containing preservatives were shorter, thinner, and had lower average weight (by 3.7–5.2%) compared to fillets without preservatives; however, these differences in physical traits were not statistically significant (Table 1). In turn, fillets with preservatives showed a significantly higher acidity (*p* < 0.05) and a slightly higher salt concentration (by 0.06 g/100 g).

An increase in the concentration of acetic acid and salt in marinated herring results in the denaturation of myofibrillar proteins such as myosin and the decrease in the pH of the muscle closer to the isoelectric point (pI), which was earlier found to reduce in the water holding capacity of herring muscle tissue [12]. For fresh Atlantic herring, the myofibrillar protein system shows an isoelectric region around pH 5.0–5.5. In acid–salt marinades, chloride binding and ionic-strength effects can shift the apparent pI toward lower values by 1 pH unit at 2–3 g/100 g NaCl, to pH 4.0–4.5 [13]. In our study, however, no statistically significant differences in pH were observed between marinades with and without preservatives. The slightly higher water content in fillets with preservatives (Table 1) may therefore be explained by lower proteolysis and reduced nitrogen loss to the brine, as described later in this section. Bykowski et al. [4] found that sorbate-treated marinated herring contained up to 7–9% more total nitrogen than untreated herring, but our results did not confirm this phenomenon (Table 1). The lipid content of marinades with preservatives was 2.9% lower by the Soxhlet method (*p* < 0.05), whereas the difference determined by the Bligh–Dyer method (–2.3%) was not statistically significant (*p* > 0.05). The lipid content determined with the Bligh–Dyer method was higher than that obtained with the Soxhlet method. The observed differences between the methods are most likely due to the limited extraction of polar lipids with petroleum ether in the Soxhlet method [14].

### 2.2. Lipid Oxidation

The raw material showed very low formation of primary (PV = 4.7 mEqO_2_/kg of lipid) and secondary (AV = 1.6) products of lipid oxidation (Figure 1A,B), which reflects that the raw material was frozen only 2 days postmortem and stored frozen for a short time under vacuum.

After marinating, the oxidation indices increased 2.5–2.7 times for PV, 8–13 times for AV, and 3–4 times for TOTOX, with the most pronounced effect observed in herring with preservatives (*p* < 0.05). Nevertheless, both marinades without and with added preservatives had low values of PV 11.3 vs. 12.3 mEqO_2_/kg of lipid (*p* < 0.05) and AV 14.6 vs. 25.0 (*p* < 0.05) (Figure 1A,B) probably because of the short-time marinating. The results calculated per 1 kg of muscle show the same trend, but the effect of preservatives is lower (Figure 1). Generally, marinades with preservatives have greater lipid oxidation indices by 9.4% for PV, 71.3% for AV, and 33.7% for TOTOX (Total Oxidation Value). After marinating, the fillets are packed with various cover-brines, limiting direct access to oxygen. However, lipid oxidation still continued, driven by the propagation of pre-formed hydroperoxides and heme-iron-catalyzed radical reactions, sustained by oxygen dissolved in tissue and cover-brine. Vinegar cover-brines, especially in rolled fillets, are less conducive to herring lipid oxidation than cover-brines made from rapeseed oil or oil-in-water emulsion since they lack an added lipid phase and dissolve much less oxygen [15]. The degree of lipid oxidation in the marinated fillets in the present study was similar to that in commercially available marinated fillets in oil cover-brine (8.5–17 mEq O_2_ ∙ kg^−1^ fat) [16]. TOTOX, which can serve as a conventional measure of lipid oxidation, ranged from 24 to 52 in the purchased marinated fillets, while it was 37.1 and 49.7 in the marinated fillets analyzed in the present study (Figure 1C). Results showed by Domiszewski et al. [16] also indicate that the addition of sodium benzoate could increase lipid oxidation in marinated fillets; however, this phenomenon did not exist in all their 11 samples. This may indicate additional factors also affect the quality of lipids in marinated fillets, such as refrigerated storage time, the composition of muscle tissue (especially endogenous antioxidant content and antioxidant peptides), the season in which the herring was caught, and other variables [17]. Lipid oxidation also increased strongly in marinades vacuum stored frozen at −18 °C for 1–5 months, especially with temperature fluctuation. Herring is rich in polyunsaturated fatty acids (PUFAs), which account for about one third of total fatty acids. PUFAs are susceptible to oxidation, especially in dark muscle, which has a high content of heme proteins such as hemoglobin and myoglobin, known to initiate the lipid oxidation process in fish [18,19,20]. Wu et al. [21] demonstrated that herring co-product fractions with higher hemoglobin content and lipoxygenase activity exhibited increased lipid oxidation, highlighting the role of these endogenous factors in oxidative stability. The herring fillets used for marination in the present study were skinless, which means that the sensitive dark muscle was no longer protected from direct oxygen access [19,20]. The addition of acetic acid lowers the pH of the meat, which can increase lipid oxidation by promoting the deoxygenation of hemoglobin through the Root and Bohr effects. Deoxyhemoglobin has been considered a better catalyst for lipid oxidation than oxyhemoglobin [18], but most importantly, it is more susceptible to autoxidation with subsequent heme loss [22,23]. On the other hand, organic acids used in marinated fillets may also protect against lipid oxidation [3]. The effect of chelation on hemoglobin-derived iron by preservatives (as weak acids) is known, but chelation efficiency decreases with pH and in the presence of salt [24]. Also, the lack of effect confirms that heme-bound iron rather than low molecular weight (LMW) iron drives lipid oxidation in uncooked herring muscle, which agrees with previous work in washed fish mince model systems [22]. With regard to sorbic acid, some studies have shown that this additive can reduce oxidation products in fish, while other studies have shown that it has no effect on the formation of oxidation-related products [8]. Here, we hypothesize that the main factor explaining a higher degree of lipid oxidation in marinades with preservatives resulted from a lower content of protein hydrolysis products, especially peptides (Table 2). Proteolysis-derived peptides can provide antioxidant effects mainly through metal chelation and to a lesser extent, radical scavenging [25]; however, their efficacy is dependent on the food matrix and may be limited in heme-mediated oxidation of fish muscle. Previous studies on herring-marinating brines demonstrated that peptide and protein fractions exert significant radical-scavenging and iron-chelating activities, protecting lipids from oxidation under acid–salt conditions comparable to those used here [26,27]. This supports the proposed link between reduced peptide pools and accelerated oxidation. In addition, other mechanisms like the redistribution or hydrolysis of pro-oxidants may also contribute. The degree of lipid oxidation in marinated fillets and other seafood is not regulated by legislation, but it has a significant impact on sensory quality and nutritional value [28]. Numerous studies have confirmed that the process of lipid oxidation in fish leads to rancid odour [19,20,23,24,29,30,31] as well as the degradation of polyunsaturated fatty acids (PUFAs) and vitamins [17,32]. Certain oxidation products are also ascribed toxic properties if consumed in high levels [27,28].

### 2.3. Nitrogen Fractions and Proteolytic Activity

Lowering the pH and increasing the salt concentration in marinated fillets resulted in the activation of selected acidic muscle proteases responsible for enzymatic ripening, i.e., hydrolysis of the muscle proteins to peptides and free amino acids [32]. The use of preservatives caused a significant (*p* < 0.05) reduction in the free amino acid fraction by 6.0% in muscle (Table 2).

Bykowski et al. [4] found that sorbate reduced the amino nitrogen content of marinated herring by 5–16%, depending on the storage time. Szymczak et al. [33] showed that the nitrogen content of marinated muscle is influenced by its losses into the marinating brine, whereby nitrogen diffusion into brine strongly depends on the ripening stage and kind of raw-material preliminary treatment. The results showed that preservatives significantly lowered the content of total nitrogen by 4.8% (*p* < 0.05), non-protein nitrogen by 5.4% (*p* < 0.05), peptides by 5.7% (*p* < 0.05), and free amino acids fractions by 7.4% (*p* < 0.05) in the marinating brine at the end of the 7 days of marination (Figure 2). The lower diffusion of nitrogen from muscle into brine therefore confirms the slower ripening of marinades with preservatives. The observed decrease in peptide and free amino acid fractions may also have implications for lipid stability. Protein and peptide fractions derived from herring-marinating brines have been shown to possess strong antioxidant properties, retarding lipid oxidation even under acidic and high-salt conditions [26]. Thus, the preservative-induced inhibition of proteolysis could indirectly enhance lipid oxidation by reducing these protective components.

The results in Table 2 further show that the addition of preservatives significantly inhibited the activity of cathepsin D, cathepsin L, and leucine-aminopeptidase (LAP) and alanine-aminopeptidase (AAP) activity in marinated fillets by 1.8% (*p* < 0.05), 3.5% (*p* < 0.05), 44.8% (*p* < 0.05), and 22% (*p* < 0.05), respectively. The very strong, ca. 97% (*p* < 0.05), reduction in aminopeptidase activity after marinating (raw material vs. marinated) is consistent with the documented sensitivity of these enzymes to acidification [34]. Thus, preservatives influenced the activity of proteases responsible for both texture (endopeptidases) and flavour (exopeptidases) development in marinated herring. A stronger inhibition of cathepsin L compared to cathepsin D may indicate that preservatives had a greater effect on reducing the hydrolysis of collagen than muscle proteins. This is important information as changes induced by collagenolytic proteases, rather than proteases that hydrolyze myofibrillar proteins, are more responsible for the hardness of marinated herring [35].

A greater inhibition of LAP than of AAP agrees with the lower loss of free amino acids and may indicate that preservatives reduced the formation of free amino acids associated with bitter flavours twice as much as those associated with sweet flavours. According to Meyer [9], the inhibition of herring proteolysis only occurs at sorbic acid concentrations higher than 0.2 g/100 g, which is twice the legally acceptable level. Here, we saw an effect below 0.1 g/100 g. Benzoate and sorbate preservatives at legal levels not only have a negative effect on the ripening of marinated fish muscle, but may also interfere with protein digestion in the gastrointestinal tract when consumed by humans [34]. Preservatives in this study can therefore promote incomplete protein digestion and reduce the release of biologically active peptides, thereby decreasing the nutritional value of such foods.

### 2.4. Colour, Texture Profile, and Sensory Evaluation of Marinated Herring Fillets

The colour parameters of the fillet surface were typical for marinated herring [33,36]. Overall, lightness increased while redness and yellowness decreased (Table 3), the latter (*p* < 0.05) significantly more without than with preservatives. The redness loss likely reflected metHb-formation. The ΔE coefficient indicates the colour difference between the tested marinated herring. The ΔE value of 2.23 in the present study is at the limit of the colour perception threshold (JND = 2.3; ASTM D2244), hence the change is likely to remain imperceptible to the average consumer [37]. Brightness (L*) increased significantly (*p* < 0.05), while changes in redness (a*) and yellowness (b*) were not significant.

Results of texture profile analyses show no significant differences between marinades with and without preservatives (Table 3), probably due to high variability in the composition of herring batches despite their grading [38]. The overall sensory acceptability of the marinated herring was very high, and the individual sensory attributes were typical of marinated fish (Figure 3).

The shape of the fillets in both samples was regular and characteristic. Despite the fact that the herring with preservatives (4.29 points) did not score significantly higher than the herring without preservatives in terms of appearance (4.11 points), assessors observed less separation/lamination between the dorsal and ventral muscles and between individual myomeres in the herring with preservatives. This confirms that preservatives reduced the activity of cathepsin L (collagenase), which breaks down the myocommata responsible for maintaining the connection between myomeres. The odour of the preservative-free marinades was typical and showed no signs of spoilage (4.63 points). In contrast, a distinct chemical odour was noticeable in the marinades with preservatives, which may have masked the desirable aroma of ripened meat (4.29 points). The sour and salty flavour of the fish marinated without preservatives was well-balanced (4.87 points), whereas fish marinated with preservatives showed a less balanced sour and sweet taste. In addition, the marinated fillets with preservatives had a chemical and bitter aftertaste, which resulted in a lower flavour score (4.51 points). The bitter taste of sorbate-marinated fillets was also reported by Bykowski et al. [4]. The taste of sorbic acid is already noticeable at a concentration of 50 mg/100 g and its degradation products negatively affect the palatability, while benzoic acid has a mild phenolic aftertaste [5]. Similarly, Fraqueza and Dias [39] showed that during the prolonged storage of marinated herring and anchovies, the addition of sodium benzoate promoted the appearance of undesirable sensory changes, particularly a metallic aftertaste, which can be a sign of lipid oxidation. In contrast, Özogul et al. [40] reported that sodium benzoate (0.5 g/100 g brine) in seafood salads had only a minimal effect on taste, and this was observed after 6 months of refrigerated storage.

In terms of texture, the fillets marinated with and without preservatives showed typical ripeness lacking discernible muscle structure, which indicates the uniform hydrolysis of structural proteins, and therefore the differences between scores (4.73 vs. 4.79) were not significant. No water was released, nor was dryness perceived during the chewing of both marinated herring samples. However, preservative-free marinated fillets were rated as juicier and less rubbery than those with preservatives, confirming the results of protease activity and contradicting the finding from the texture profile analysis. The mean sensory scores showed that overall sensory acceptability was higher for marinated fillets without preservatives (4.60 points) compared to those with preservatives (4.45 points; *p* > 0.05; Figure 3). Despite the small difference of 0.15 points (4.60 − 4.45 = 0.15) and lack of significance at *n* = 3, Cohen’s d value is 0.6, which suggests a moderate perceptual difference, requiring a larger panel to confirm.

Although the sensory panel consisted of only seven trained assessors, all had extensive experience in evaluating marinated fish products. This small panel size may have limited the statistical power to detect subtle differences among sensory attributes. Nevertheless, the consistent identification of a mild chemical odour and aftertaste by all panellists indicates that this sensory change was real and perceptible, even though the overall flavour score did not differ significantly. This finding illustrates that preservatives can introduce slight off-notes without substantially affecting the overall sensory acceptability, as the mean flavour score represents an average of multiple attributes, only one or two of which (chemical or bitter notes) were negatively affected.

### 2.5. Microbiological Analysis

The results of the microbiological analysis confirmed the high microbiological quality of the raw herring. The herring blocks were vacuum packed with seawater added, allowing the TVC index to start at 50 CFU/g despite slow thawing (Table 4).

The increase in the number of psychrophilic bacteria in the marinated fillets without preservatives had two orders of magnitude, reaching a level of 12 × 10^2^ CFU/g, and yeast was also detected at 9 × 10^2^ CFU/g, probably due to cross-contamination in the production plant (Table 4). In marinated herring, most psychrophilic bacteria are usually derived from lactic acid bacteria [41]. The a_w_ value was determined to be 0.978 ± 0.012, which is within the typical range of herring marinades (0.96–0.98). The marinated fillets with added preservatives had no detectable levels of any of the microorganisms tested, confirming the effectiveness of the preservatives in the acidic environment of the marinades. The preservatives can inhibit the activity of enzymes, especially dehydrogenases, as well as yeasts and moulds, but were equally effective in eliminating *Enterobacteriaceae*, *Bacillus cereus*, *Listeria monocytogenes*, and *Staphylococcus aureus*. The effect of potassium sorbate (E 202) is limited to catalase-positive bacteria, and mould multiplication is inhibited at concentrations as low as 0.07–0.1 g/100 g. Sodium benzoate, on the other hand, has a strong effect on yeasts and bacteria but a weak impact on moulds and lactic acid bacteria [5]. The antimicrobial activity of both preservatives is strongest in an acidic environment due to their low pKa-values, with at least 60–75% of the added preservatives being in an undissociated (active) form at a pH close to 4.0, which is typical for marinated fish.

Overall, the use of preservatives makes it possible to reduce the total number of psychrophilic bacteria, yeasts, and moulds in marinated fillets, thus maintaining the intended 2–6 months shelf life of the mildly marinated herring fillets. In a previous study, sodium benzoate in weakly salted herring fillets reduced the diversity of bacteria and yeast but had no effect on the total number of bacteria and yeasts, as the empty ecological niche was replaced by other bacteria [40]. Further, Bykowski et al. [4] found that sorbate did not improve the microbiological quality of marinated fish and at the same time, the authors pointed out that the raw material and the hygienic conditions during the marinated fish production had a larger effect on the microbial quality. Bykowski et al. [4] also reported that the inhibition of one microbial group (e.g., lactic acid bacteria) by preservatives may allow others to occupy the ecological niche. In our study, marinades treated with preservatives had bacterial counts below 10 CFU/g, which does not allow for meaningful conclusions to be drawn regarding changes in the microbial community after 7 days. Although sub-detectable or VBNC (viable but non-culturable) populations cannot be ruled out, potential ecological replacement phenomena would be more relevant during long-term storage after packaging and were beyond the scope of this study. Additionally, high concentrations of salt and acetic acid do not always inhibit the proliferation of lactic acid bacteria, moulds, and yeasts in marinated fish, which are able to continue their activity more or less rapidly depending on adaptation [42]. As stated initially, bacterial growth in marinated fish also depends on the type of cover-brine—with lower growth seen in fillets with acetic acid brine than with oil or sauce [15]. Thus, preservatives still play an important role for the microbial quality of herring marinades.

## 3. Materials and Methods

### 3.1. Marinated Herring Fillets

Atlantic herring (*Clupea harengus* L.) fillets without skin, frozen 4 months in 20 kg vacuum-packed blocks with 2 kg of vacuum-brine (seawater) were thawed at 21 °C in 12 h (Defrosting chamber 12s, Revic, Poland) from −18 °C to +2 °C in the centre of the block. Then, 200 kg thawed fillets were marinated for 7 days at 4 ± 1 °C in 200 kg brine using a tank of 400 L. The brine consisted of 904.5 g/kg water, 60 g/kg sodium chloride (vacuum salt), 32 g/kg acetic acid (80 g/100 g vinegar essence), 2.5 g/kg sodium benzoate, and 1 g/kg potassium sorbate. Control fillets were marinated in brine without preservatives. Both samples were prepared in an industrial scale from the same batch of herring fillets using the same processing parameters in triplicate in separate tanks. The procedure reflects standard industrial marination practice and was carried out in cooperation with a commercial herring processing facility. During the first 6, 12, and 48 h of marinating, fillets were mixed manually in the brine. After 7 days of marinating, 30 fillets were taken at randomly from the centre of each tank and brought to the laboratory together with the brine. Sampling and transportation took 2 h. All fillets were graded in terms of weight and length with an accuracy of 0.1 cm and 0.1 g, respectively. Nine uniform fillets were randomly selected for sensory and four for textural analyses. The remaining marinated fillets or thawed raw materials fillets were minced to a uniform consistency and subjected to analyses.

### 3.2. Total Acidity, pH, Salt Content, and Moisture Content

The pH was determined using a digital pH metre (F20, Mettler Toledo, Columbus, OH, USA) in water extract (1:5, *w*:*v*). The content of moisture in the muscle was measured by a gravimetric method after drying at 105 °C (No. 950.46B), total acidity was measured by a titration method (No. 935.57), and salt by the Mohr titration method (No. 937.09). All methods were standard AOAC analytical methods, and measurements were performed in triplicate on each sample.

### 3.3. Lipid Content and Oxidation Index

Total lipid content in the muscle was determined using the Soxhlet method (AOAC No. 960.39) with petroleum ether extraction, and by the Bligh–Dyer method [43] with chloroform–methanol extraction (2:1, *v*:*v*). Also, in the Bligh–Dyer extracts, primary and secondary lipid peroxidation products were determined. The peroxide value (PV) was determined on the chloroform phase with the ferric thiocyanate technique to allow peroxide reduction and red ferric complex formation, which were measured at 470 nm [44], and the results were expressed as mEqO_2_/kg of lipid and as mEqO_2_/kg of muscle. The anisidine value (AV) was also determined on the chloroform phase, based on the reaction between α- and β-unsaturated aldehydes (primarily 2-alkenals) and p-anisidine reagent. Total Oxidation Value (TOTOX) was calculated using the values determined for peroxide and p-anisidine (2PV + AV). Measurements were performed in triplicate.

### 3.4. Total Protein and Non-Protein Nitrogen Content

Total nitrogen was determined by the Kjeldahl method (AOAC No. 940.25) and protein content was estimated using a nitrogen-to-protein conversion factor of 5.58. In 5 g/100 g TCA extracts of the muscle: (i) non-protein nitrogen (NPN) was determined by the Kjeldahl method, and (ii) protein hydrolysis products of the peptides fraction (PHP(R)) and the free amino acids fraction (PHP(A)) were determined by the Lowry method, modified by Kołakowski [45]. Measurements were performed in triplicate.

### 3.5. Cathepsins and Aminopeptidase Activity

The activities of cathepsins D, B, and L, as well as aminopeptidases, were determined in the supernatants of aqueous extracts of marinated meat (10:1, *v*:*m*) [36]. Cathepsin D activity was measured at pH 4.5 against MCA-Gly-Lys-Pro-Ile-Leu-Phe-Phe-Arg-Leu-Lys-(DNP)-D-Arg-NH_2_ as the substrate with and without pepstatin-A. Cathepsin B and B+L activities at pH 5.5 were determined against Z-Arg-Arg-MCA and Z-Phe-Arg-MCA as substrates, respectively. Cathepsin L activity was calculated as the difference between the activities of cathepsin B+L and cathepsin B activity. One unit (U) of cathepsin activity was expressed in fluorescence unit increase per minute per gram of muscle at 37 °C. Leucine-aminopeptidase and alanine-aminopeptidase activities at pH 7.0 were assessed using specific chromogenic substrates, Leu-pNa and Ala-pNa, respectively. One unit (U) of aminopeptidase activity was expressed as 1 nM pNa per minute per gram of muscle at 37 °C [36]. Measurements were performed in duplicate.

### 3.6. Texture Profile Analyses (TPA) and Colour of Fillets

The texture profile of the dorsal muscle of fillets was analyzed using a TA.XT plus texture analyzer (Stable Micro Systems, Godalming, UK). Hardness, adhesiveness, springiness, cohesiveness, gumminess, chewiness, and elasticity of the muscle were determined [33]. A double compression test was performed using a flat cylindrical P10 probe (10 mm diameter) up to 50% deformation of the fillet height, at a speed of 5 mm/s with a relaxation time of 5 s. The determination was performed at 3 dorsal muscle sites for 4 fillets from each sample (*n* = 12).

The colour measurement of the fillet surface (dorsal muscle) was performed instrumentally using a colorimeter (WR 18 FRU, Shenzhen Wave Optoelectronics Technology Co, Ltd., Shenzhen, China) based on a white reference plate (L* = 92.4; a* = −0.04; b* = +1.9) and the CIELab system, illuminant D65, 10° observer, illumination mode d/8 with an 8 mm aperture. The following colour coordinates were determined: L* (brightness), a* (redness), and b* (yellowness). The colour difference (∆E = [(∆L)^2^ + (∆a)^2^ + (∆b)^2^]^0.5^) was calculated for the mean lab values where ∆L, ∆a, and ∆b are the differences in L*, a*, and b* values between marinades with and without preservatives, respectively [33]. Texture and colour analyses of fillets were performed in 12 replicates.

### 3.7. Sensory Evaluation

Sensory analysis of the samples was based on the method described by Kamiński et al. [36]. The evaluation was performed by a panel of 7 members (3 men and 4 women) trained in the sensory analysis of fish marinades [46]. Three fillets from each sample were served on porcelain plates. Each panellist was given three pieces (each 2–3 cm wide), one from each fillet. The evaluated area corresponded to between 2/10 and 6/10 of the fillet length measured from the head side. The evaluation was conducted in individual sensory booths, free from foreign odours, in daylight and at room temperature. Panellists used water and white bread to clean their palates between tastings. Sensory attributes such as appearance, taste, aroma, and texture [47] were assessed on a 5-point hedonic scale, with the possibility of half-point ratings (0 indicating the lowest score, and 5 the highest). The overall sensory acceptability was represented by the average score of the attributes examined. The effect size of the difference independent of the number of trials (Cohen’s d) was calculated from the formula: d = (mean1 − mean2)/SD_pooled_.

### 3.8. Microbiology Analyses

Samples in triplicate were weighed at 20 g each into sterile stomacher bags and 180 mL of sterile dilution fluid (P-0061, BTL, Warsaw, Poland) was added. Samples were homogenized (BagMixer 400P, Interscience, Saint Nom la Bretêche, France) and then dilutions and surface cultures were performed on a nutrient agar medium (P-0075, BTL, Poland) for total psychrophilic bacteria, on nutrient agar (01140, Scharlab, Barcelona, Spain) for total mesophilic bacteria [48,49], on Sabouraud medium for total yeast and mould [50], and on MYP Agar for *Bacillus cereus* [51]. Psychrophilic and mesophilic counts were distinguished by incubation temperature (7 °C vs. 30 °C, respectively). In addition, flooded cultures were performed on VRBGLA medium (BT5158.02, Biomaxima, Lublin, Poland) for *Enterobacteriacea* [52], on LSA and ALOA for *Listeria monocytogenes* [53], on XLD and BGA for *Salmonella* [54] and on Baird-Parker for *Staphylococcus aureus* [55]. The limit of quantification of the plate method was 1 log CFU/g; results were recorded as <10 CFU/g. Water activity (a_w_) in marinades was determined using HydroLab device (Rotronic AG, Bassersdorf, Switzerland).

### 3.9. Statistical Analyses

Statistical analysis was performed using Statistica 13.1 (Statsoft, Tulsa, OK, USA). After Student’s *t*-test, mean values and standard deviations were calculated. Statistically significant differences between means were calculated using Tukey’s post hoc test (*p* < 0.05). Treatment was set as a fixed factor, tank as a random factor (30 fillets nested within each tank). Results in tables and figures are show as mean ± SD; mean value represents 30 fillets taken from one tank, N = 3 independent tanks.

## 4. Conclusions

The addition of preservatives had little effect on the sensory quality of the marinated fillets, and the difference may not be noticeable to the consumer. The most pronounced effects were a nicer appearance, but a chemical and bitter aftertaste. However, the use of preservatives significantly decreased the content of lipids, free amino acids, and peptides in muscle, losses of nitrogen compounds to brine, as well as cathepsins and amino-peptidase activity of the tested fillets. At the same time, it was also found that preservatives may be necessary in production facilities where microbial cross-contamination occurs; marinated herring fillets without preservatives showed yeast and psychrophiles at 2 log. The present study focused on the 7-day marination phase, corresponding to the industrial production step before packing in cover-brines or sauces. This period captures the most intense biochemical changes, including protease inhibition and the onset of lipid oxidation. Although additional processing, longer storage could further modify these effects, particularly under different cover-brine conditions, extending the experimental duration would have introduced multiple additional variables unrelated to the direct action of preservatives.

Although the preservatives are not known as pro-oxidants per se, marinades where they were added had significantly higher lipid oxidation indices than marinades without preservatives. Taking into account the observed reduction in the peptide fraction along with numerous reports of the antioxidant activity of peptides in marinated herring, the observed increases in PV and AV may be related to a reduction in antioxidative peptides due to the inhibition of proteolysis. Although the use of preservatives had negative effects on protease activity and lipid stability, it effectively reduced microbial growth and nutrient loss from the muscle to the surrounding brine. Therefore, their inclusion remains justified in processing environments where cross-contamination or insufficient hygiene may compromise product safety. A rational, minimal-use strategy is recommended; preservatives should be applied only at levels necessary to ensure microbiological safety, while avoiding the excessive inhibition of proteolysis and associated sensory or nutritional deterioration. Further studies should focus on other preservatives not hampering proteolysis and stimulating lipid oxidation in order to maximize both the safety and eating quality of marinated herring products.

## Figures and Tables

**Figure 1 molecules-30-04103-f001:**
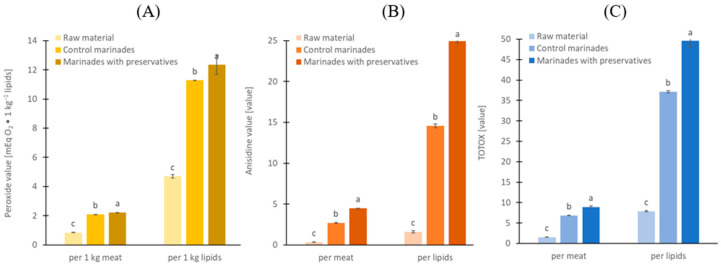
Lipid oxidation indicators: (**A**) peroxide value, (**B**) anisidine values, and (**C**) TOTOX, expressed per marinated meat and per lipids; ^a, b, c^ results with the same letter do not differ significantly (*p* > 0.05).

**Figure 2 molecules-30-04103-f002:**
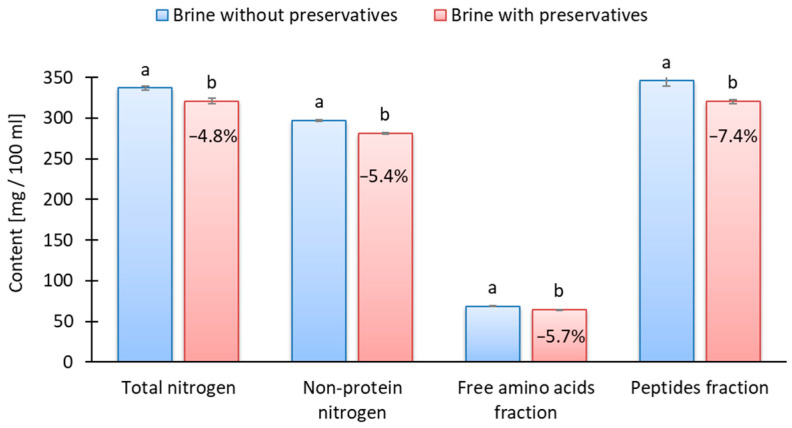
The effect of preservatives addition on total and non-protein nitrogen fractions of marinating brine; ^a, b^ results with the same letter do not differ significantly (*p* > 0.05); value given on the red bar shows the reduction in fraction content compared to the sample without preservatives.

**Figure 3 molecules-30-04103-f003:**
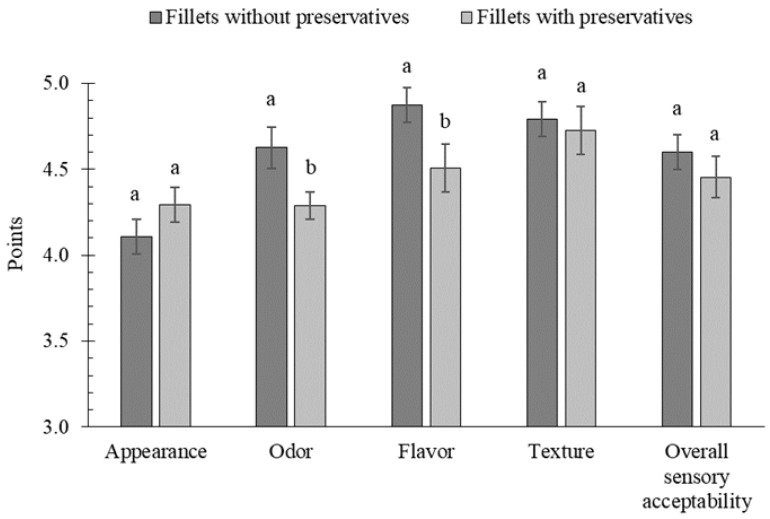
The effect of preservative addition on sensory attributes and overall sensory acceptability of marinated fillets; ^a, b^ results with the same letter do not differ significantly (*p* > 0.05).

**Table 1 molecules-30-04103-t001:** The effect of the addition of preservatives on fillet size and the proximate composition of marinated muscle; differences were calculated between samples with and without preservatives.

Analysis	Raw Fillet	Marinades—No Preservatives	Marinades—With Preservatives	Difference [%]
**Physical properties:**				
Mass of fillets [g]	60.3 ± 3.5 ^a,^*	54.3 ± 2.9 ^b^	51.5 ± 4.5 ^b^	−5.2
Length of fillets [cm]	14.6 ± 1.1 ^a^	15.6 ± 1.4 ^a^	14.9 ± 0.9 ^a^	−4.5
Fillet thickness [mm]	14.21 ± 0.54 ^a^	12.26 ± 0.97 ^b^	11.81 ± 0.74 ^b^	−3.7
pH value	6.65 ± 0.01 ^a^	3.98 ± 0.03 ^b^	3.97 ± 0.03 ^b^	−0.2
**Proximate composition:**				
Total acidity [g/100 g]	0.22 ± 0.05 ^c^	1.78 ± 0.02 ^b^	1.84 ± 0.01 ^a^	3.4
Salt [g/100 g]	0.20 ± 0.02 ^b^	2.90 ± 0.03 ^a^	2.96 ± 0.02 ^a^	2.1
Moisture [g/100 g]	67.2 ± 0.1 ^a^	60.6 ± 0.06 ^c^	60.8 ± 0.06 ^b^	0.3
Lipids by Soxhlet [g/100 g]	12.8 ± 0.06 ^c^	17.0 ± 0.02 ^a^	16.5 ± 0.1 ^b^	−2.9
Lipids by Bligh–Dyer [g/100 g]	13.9 ± 0.26 ^b^	18.4 ± 0.42 ^a^	18.0 ± 0.23 ^a^	−2.3
Crude Protein [g/100 g]	12.95 ± 0.1 ^a^	13.05 ± 0.22 ^a^	13.33 ± 0.12 ^a^	2.1

* ^a, b, c^ Averages in the same row indicated by the same lower-case letter are not significantly different.

**Table 2 molecules-30-04103-t002:** The effect of preservative addition on non-protein nitrogen content (NPN), free amino acids (FAAs), peptides, and protease activity of marinated herring fillets; differences were calculated between samples with and without preservatives.

Analysis	Raw Fillet	Marinades—No Preservatives	Marinades—With Preservatives	Difference [%]
NPN [mg/100 g]	235 ± 8.2 ^a,^*	254.2 ± 13.0 ^a^	246.9 ± 8.3 ^a^	−2.9
FAA [mg/100 g]	22.3 ± 1.1 ^c^	78.0 ± 1.9 ^a^	73.3 ± 0.9 ^b^	−6.0
Peptides [mg/100 g]	56 ± 3.2 ^c^	604.3 ± 6.2 ^a^	557.2 ± 7.3 ^b^	−8.8
Cathepsin D [U]	2887 ± 32 ^c^	7228 ± 20 ^a^	7099 ± 24 ^b^	−1.8
Cathepsin B [U]	1504 ± 17 ^a^	805.0 ± 11 ^b^	786.0 ± 6.0 ^c^	−2.4
Cathepsin L [U]	487 ± 11 ^c^	1420 ± 21 ^a^	1370 ± 22 ^b^	−3.5
LAP [U]	153 ± 6.1 ^a^	4.22 ± 0.3 ^b^	2.33 ± 0.3 ^c^	−44.8
AAP [U]	114 ± 3.4 ^a^	3.0 ± 0.2 ^b^	2.34 ± 0.3 ^c^	−22

* ^a, b, c^ Averages in the same row indicated by the same lower-case letter are not statistically significantly different, LAP—lue-aminopeptidase; AAP—ala-aminopeptidase.

**Table 3 molecules-30-04103-t003:** The effect of preservative addition on surface colour and texture profile of marinated herring fillets; differences were calculated between samples with and without preservatives.

Analysis	Raw Fillet	Marinades—No Preservatives	Marinades—With Preservatives	Difference [%]
**Colour of fillets’ surface:**				
L* (brightness)	61.25 ± 0.87 ^b,^*	67.66 ± 1.08 ^a^	65.57 ± 1.15 ^a^	−3.1
a* (redness)	4.30 ± 0.32 ^a^	3.21 ± 0.40 ^b^	3.21 ± 0.39 ^b^	0.0
b* (yellowness)	3.54 ± 0.18 ^a^	2.24 ± 0.26 ^b^	3.03 ± 0.37 ^a^	26.1
Colour difference (ΔE) between marinades:				2.23 unit
**Texture profile analysis:**				
Hardness [N]	9.12 ± 1.29 ^a^	8.13 ± 1.09 ^a^	7.39 ± 1.01 ^a^	−9.1
Adhesiveness [mJ]	4.56 ± 0.90 ^a^	4.01 ± 0.99 ^a^	3.66 ± 0.88 ^a^	8.7
Springiness [m]	2.12 ± 0.11 ^a^	0.82 ± 0.06 ^b^	0.81 ± 0.06 ^b^	−1.2
Cohesiveness [-]	0.80 ± 0.04 ^a^	0.51 ± 0.03 ^b^	0.48 ± 0.04 ^b^	−5.9
Gumminess [N]	6.21 ± 0.42 ^a^	4.25 ± 0.61 ^b^	3.38 ± 0.45 ^b^	−20.5
Chewiness [J]	4.11 ± 0.60 ^a^	3.6 ± 0.68 ^ab^	2.71 ± 0.48 ^b^	−24.7
Resilience [N]	0.235 ± 0.007 ^a^	0.153 ± 0.011 ^b^	0.135 ± 0.009 ^b^	−11.8

* ^a, b^ Averages in the same row indicated by the same lower-case letter are not statistically significantly different.

**Table 4 molecules-30-04103-t004:** The effect of preservatives on microbial count in herring fillet raw material and in fillets marinated for 7 days (CFU/g).

Analyses	Herring Fillet Raw Material	Fillets Without Preservatives	Fillets with Preservatives
Psychrophiles	5 × 10^1^	12 × 10^2^	<10
Mesophiles	<10	<10	<10
Yeasts	<10	9 × 10^2^	<10
Moulds	<10	<10	<10
*L. monocytogenes*	absent	absent	absent
*S. aureus*	<10	<10	<10
*Salmonella*	absent	absent	absent
*B. cereus*	<10	<10	<10
*Enterobacteriaceae*	<10	<10	<10

## Data Availability

Data are contained within the article.

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
