# Peer review of "Sodium Benzoate and Potassium Sorbate Inhibit Proteolysis and Promote Lipid Oxidation in Atlantic Herring Marinades Produced on an Industrial Scale"

_molecules, 2025, doi:10.3390/molecules30204103_

Round 1

Reviewer 1 Report

Comments and Suggestions for Authors
  1. Line 28, TOTOX value only has its abbreviation written. Its full name should be indicated when it first appears.
  2. Why the texture and color results was not mentioned in the abstract section?
  3. Line 49, what is the inhibitory effect of sodium benzoate and potassium sorbate on L. monocytogenes? What is the mechanism for sodium benzoate and potassium sorbate extend the shelf life of marinades herring?
  1. In this study, the title and results described the effect of Sodium benzoate and potassium sorbate on lipid oxidation. In the introduction section, it should supplement the  motivation for studying the antioxidant effects of these two substance.
  2. Line 94, EFSA only has its abbreviation written. Its full name should be indicated when it first appears.
  3. Line 109-112, as the observed differences were not statistically significant, so the shorter expressionin Line 109 and the higher expression in Line 110 higher is not statistically significant. In addition, it is better to describes the results of Physical properties and proximate composition separately in 2.1 section.
  1. Line 122-123, “This likely explains why in the present study lower pH of herring marinated with preservatives significantly raised the water...”. The author explained the results of pH. In fact, there was no difference in pH, it should be checked carefully.
  2. Line 142, it write as increased several-fold, please accurately describe the multiple of the increase.
  3. In the section of “1. Marinated herring fillets”, the method of Marinated herring fillets is unique for your laboratory or it was a general processing methods, please provide some references.
  4. The method used in this study should provide some references, such as “3. Lipid content and oxidation index”, “3.4. Total protein and non-protein nitrogen content”and “3.5. Cathepsins and aminopeptidases activity”
  5. Line 407, the “2”in the formula (∆E = [(∆L)2 + (∆a)2 + (∆b)2]0.5 should be superscript 
  6. Line 429, the “nutrient agar medium”(P-0075, BTL, Poland) for total psychrophilic bacteria and the “nutrient agar” (01140, Scharlab, Barcelona, Spain) for total mesophilic bacteria is the same agar? What is the difference between them?
  7. In the entire result and discussion section, whether the result is significant should be accurately stated. After describing a certain result, it should be directly discussed.
  8. The references cited in this manuscript are too old. Among the 45 references, only 8 references were published in the lately five years. More newly study should be cited, such as the lately five years.

Author Response

Dear Reviewer 1, 

We would like to express our sincere gratitude for your thorough review of our manuscript and for the insightful comments and constructive suggestions you have provided. Your feedback has been extremely valuable in helping us improve the overall quality, clarity, and rigor of the paper. 

In this revised version of the manuscript, we have carefully considered each of your remarks. Wherever possible, we have incorporated the recommended changes, and we believe that these revisions have strengthened the paper. For the sake of transparency, our responses are presented point by point, corresponding to each of the reviewers’ comments. In cases where we felt it appropriate to maintain our original approach, we have provided detailed justifications. 

We hope that the revisions adequately address your concerns and that the improved manuscript meets the high standards expected by the journal. We are truly grateful for the opportunity to revise our work and for your valuable contribution to enhancing its quality. 

Comment 1: Line 28, TOTOX value only has its abbreviation written. Its full name should be indicated when it first appears. 

Response 1:  We thank the Reviewer for this helpful remark. In the revised manuscript we have now provided the full name of the TOTOX index when it first appears in the Results section.
It is now written as Total Oxidation Value (TOTOX), with the abbreviation indicated in parentheses.

Comment 2: Why the texture and color results was not mentioned in the abstract section? 

Response 2:  We appreciate the Reviewer’s comment. In the Abstract, we aimed to highlight the most pronounced and statistically significant effects of the preservatives, namely the inhibition of proteolysis and the promotion of lipid oxidation. Texture and colour changes were analysed; however, the observed differences between samples with and without preservatives were not statistically significant, as reported in the Results section. Therefore, to maintain clarity and conciseness in the Abstract, these results were not included.

Comment 3: Line 49, what is the inhibitory effect of sodium benzoate and potassium sorbate on L. monocytogenes? What is the mechanism for sodium benzoate and potassium sorbate extend the shelf life of marinades herring? 

Response 3:  We thank the Reviewer for raising this point. The antimicrobial activity of sodium benzoate and potassium sorbate is primarily associated with their undissociated forms at acidic pH, which can penetrate microbial cell membranes, leading to intracellular acidification, inhibition of dehydrogenase enzymes, and disruption of metabolic processes. This mechanism is particularly relevant for Listeria monocytogenes and other acid-sensitive microorganisms, thereby contributing to the extended shelf life of marinated herring. In the revised manuscript, we have added a clarifying sentence in the Introduction (Line 50).

Comment 4: In this study, the title and results described the effect of Sodium benzoate and potassium sorbate on lipid oxidation. In the introduction section, it should supplement the  motivation for studying the antioxidant effects of these two substance. 

Response 4:  We appreciate the Reviewer’s observation. However, our study did not aim to investigate sodium benzoate and potassium sorbate as antioxidant compounds per se, as these preservatives are not generally recognised to possess intrinsic antioxidant activity. Their only potential contribution in this regard could be associated with weak chelation of iron ions. Rather, our motivation was to examine whether their inhibition of proteolysis, and the consequent reduction in the release of antioxidant peptides, could indirectly influence lipid oxidation in marinated herring. This rationale was already introduced in the final paragraph of the Introduction (Lines 88–89). We therefore consider the current framing of the study motivation to be appropriate, although we have slightly clarified the text to minimise the risk of misinterpretation.

Comment 5: Line 94, EFSA only has its abbreviation written. Its full name should be indicated when it first appears. 

Response 5:  We thank the Reviewer for this remark. In the revised manuscript, the full name European Food Safety Authority (EFSA) has been provided  when it first appears (Line 94).

Comments 6: Line 109-112, as the observed differences were not statistically significant, so the shorter expressionin Line 109 and the higher expression in Line 110 higher is not statistically significant. In addition, it is better to describes the results of Physical properties and proximate composition separately in 2.1 section. 

Response 6:  We thank the Reviewer for this comment. We fully agree that the differences in fillet length and salt content were not statistically significant, and we have clarified this point in the revised text. Nevertheless, we consider it important to report these observations because, even if not significant at p < 0.05, they may help explain related technological phenomena (e.g., water-holding capacity, proteolysis rate) and could be of practical relevance for fish processors. For this reason, we have decided to retain these data, but we have rephrased the sentences to avoid any misinterpretation. Following the Reviewer’s advice, we also restructured Section 2.1 so that physical properties and proximate composition are described separately for greater clarity.

Comment 7: Line 122-123, “This likely explains why in the present study lower pH of herring marinated with preservatives significantly raised the water...”. The author explained the results of pH. In fact, there was no difference in pH, it should be checked carefully. 

Response 7: We thank the Reviewer for pointing out this inconsistency. Indeed, the pH values between marinades with and without preservatives did not differ significantly, whereas  the difference in total acidity was statistically significant. In the revised manuscript, we have corrected this by removing the explanation that linked water-holding capacity directly linked to pH. Instead, we now emphasize that the small but significant changes in acidity and proteolysis may account for  the slightly higher water content observed in fillets containing preservatives.

Comment 8: Line 142, it writes as increased several-fold, please accurately describe the multiple of the increase. 

Response 8:  We thank the Reviewer for this helpful  suggestion. In the revised manuscript, we have replaced the vague expression “increased several-fold” with the exact multiples of increase relative to the raw material values: the peroxide value increased 2.5–2.7 times, the anisidine value increased 8–13 times, and the TOTOX value increased 3–4 times (Fig. 1).

Comment 9: In the section of “1. Marinated herring fillets”, the method of Marinated herring fillets is unique for your laboratory or it was a general processing method, please provide some references. 

Response 9:  We thank the Reviewer for this insightful question. The marinating process applied in this study was not a unique laboratory method but was conducted under industrial-scale conditions, using standard technology commonly employed by fish processors. The aim was to reproduce  actual production practices as closely as possible. Because industrial procedures differ between plants (e.g.,  tank size, mixing protocols, and brine composition), we did not cite a single reference that would exactly match our conditions. Instead, we provided  the process in detail to ensure reproducibility. To clarify this point, we have revised Section 3.1 to explicitly stating that the process reflects common industrial practice rather than a laboratory-specific method.

Comment 10: The method used in this study should provide some references, such as “3. Lipid content and oxidation index”, “3.4. Total protein and non-protein nitrogen content”and “3.5. Cathepsins and aminopeptidases activity” 

Response 10:  We thank the Reviewer for this comment. The methods for lipid, protein, and enzyme analyses in our study already include appropriate references. Specifically, lipid extraction and oxidation indices are referenced to AOAC (No. 960.39) and ISO 6885:2008 standards, and [41] reference for oxidation indicators; while protein and nitrogen fractions follow AOAC No. 940.25 and Kołakowski (2005). The determination of cathepsin and aminopeptidase activities was described in detail and referenced to Kamiński et al. [35], which is a well-established method for fish muscle proteases. These sources are the most relevant and widely accepted for the type of analyses conducted.

Comment 11: Line 407, the “2”in the formula (∆E = [(∆L)2 + (∆a)2 + (∆b)2]0.5 should be superscript  

Response 11: Thank you for pointing this out; the correction has been made in the revised manuscript.

Comment 12: Line 429, the “nutrient agar medium”(P-0075, BTL, Poland) for total psychrophilic bacteria and the “nutrient agar” (01140, Scharlab, Barcelona, Spain) for total mesophilic bacteria is the same agar? What is the difference between them? 

Response 12: We thank the Reviewer for this valuable observation. Both media are indeed standard nutrient agar formulations, used here from two different suppliers. The difference was not related to their composition but to logistical aspects of laboratory supply. The determination of psychrophilic and mesophilic bacteria was based on the incubation temperature (7°C vs. 30°C), rather than on the composition of the medium itself. To avoid confusion, in the revised manuscript we have clarified that both tests were conducted on standard nutrient agar, with different incubation temperatures applied for psychrophilic and mesophilic counts.

Comment 13: In the entire result and discussion section, whether the result is significant should be accurately stated. After describing a certain result, it should be directly discussed. 

Response 13: We thank the Reviewer for this suggestion. In the revised manuscript we carefully checked all sections to ensure that statistical significance is reported consistently. In the Results and Discussion, we now explicitly distinguish between significant (p < 0.05) and non-significant differences in the text, while avoiding unnecessary repetition, since the exact p-value threshold is already specified in the Materials and Methods section and significant differences are clearly indicated in tables and figures. In addition, non-significant results are now described more cautiously, to prevent  misinterpretation.

Comment 14: The references cited in this manuscript are too old. Among the 45 references, only 8 references were published in the lately five years. More newly study should be cited, such as the lately five years.

Response 14: We thank the Reviewer for this remark. We agree that including recent studies is important whenever available. However, in the specific field of preservatives in marinated herring, very few new publications have appeared in the last two decades, and most of the key works still cited in the literature date back to earlier years. For this reason, we retained these classical references, as they remain the most relevant for explaining the observed phenomena. At the same time, we have incorporated recent studies on related aspects (e.g., lipid oxidation, protease activity, and antimicrobial effects) where appropriate. We would also like to note that other reviewers explicitly recognized the lack of recent studies in this area and emphasized that our manuscript is valuable precisely because it revisits this neglected but important topic and provides novel data.

Comment 15: The English could be improved to more clearly express the research.

Response 15: We thank the Reviewer for this observation. The entire manuscript has been carefully re-edited to improve clarity and flow, with sentences shortened and wording adjusted to achieve a more natural English style typical for scientific writing.

Reviewer 2 Report

Comments and Suggestions for Authors

The paper has a fairly practical orientation.  It is well structured, understandable, and the text flows well.

 I would just like to make a few comments:

Abstract

Line: 26: It would be better to clarify that these percentages refer to brine and not to tissue.

Introduction

Line: 94: EFSA is not a legislative body but an advisory one. EU legislation is enforced through Regulations (E.E. Regulations). In this specific case, it is EU Regulation 1333/2008. The reference should be amended accordingly.

Results

Line 101 -107: "This suggests that the concentrations of sodium benzoate and potassium sorbate in 100 g of the herring muscle tested in the present study were at least 100 µg and 50 µg, respectively."

This means that the concentrations were 1 and 0.5 ppm! Are they too low?

Lines 128-132: The sentence should probably be reworded because P<0.05 does not apply in the case of the Bligh–Dyer method.

Lines 203-204: The sentence should be reworded because the reduction in FAA is 6% according to Table 2/line 2.

Lines 189-191: "heme" instead of "heam"

General comment – question: The authors believe that the reduction of natural microflora through the application of preservatives may have had an effect on reduced proteolysis. In other words, the extracellular enzymes of the bacterial microflora have no effect on the final physicochemical characteristics of the product. In other words, the extracellular enzymes of the bacterial microflora have no effect on the final physicochemical characteristics of the product. I am not asking for further experiments, but if there is relevant literature, it would be good to add it.

Author Response

Dear Reviewer 2, 

We would like to express our sincere gratitude for your thorough review of our manuscript and for the insightful comments and constructive suggestions you have provided. Your feedback has been extremely valuable in helping us improve the overall quality, clarity, and rigor of the paper. 

In this revised version of the manuscript, we have carefully considered each of your remarks. Wherever possible, we have incorporated the recommended changes, and we believe that these revisions have substantially strengthened the paper. For the sake of transparency, our responses are presented point by point, corresponding to each of the reviewers’ comments. In cases where we felt it appropriate to maintain our original approach, we have provided detailed justifications. 

We hope that the revisions adequately address your concerns and that the improved manuscript meets the high standards expected by the journal. We are truly grateful for the opportunity to revise our work and for your valuable contribution to enhancing its quality. 

Comment 1: The paper has a fairly practical orientation.  It is well structured, understandable, and the text flows well.

Response 1:  We sincerely thank the Reviewer for the positive evaluation and appreciation of the manuscript’s structure, clarity, and practical orientation.

I would just like to make a few comments:

Abstract

Comment 2Line: 26: It would be better to clarify that these percentages refer to brine and not to tissue.

Response 2:  We thank the Reviewer for this accurate observation. We have revised  the sentence to clearly indicate that the reported decreases in amino acid and peptide fractions, as well as enzyme activities, refer specifically to muscle tissue. In addition, we supplemented the Abstract with a brief mention of nitrogen fraction changes in the marinating brine, which further supports the conclusion that preservatives inhibit the ripening process.

Introduction

Comment 3Line: 94: EFSA is not a legislative body but an advisory one. EU legislation is enforced through Regulations (E.E. Regulations). In this specific case, it is EU Regulation 1333/2008. The reference should be amended accordingly.

Response 3:  We thank the Reviewer for this valuable clarification. The text has been revised accordingly: EFSA is now cited as the scientific authority responsible for assessment and safety evaluation, while the legal framework for the use of preservatives is correctly referenced to EU Regulation (EC) No 1333/2008.

Results

Comment 4Line 101 -107: "This suggests that the concentrations of sodium benzoate and potassium sorbate in 100 g of the herring muscle tested in the present study were at least 100 µg and 50 µg, respectively."

This means that the concentrations were 1 and 0.5 ppm! Are they too low?

Response 4:  We thank the Reviewer for noticing this inconsistency. The original estimate was indeed incorrect because of a unit conversion error. Based on the 1:1 fish-to-brine ratio and an average 50% transfer of preservatives from brine to muscle, the actual concentrations are approximately 1250 ppm of sodium benzoate and 500 ppm of potassium sorbate. The text has been corrected accordingly to reflect more realistic values consistent with published data for industrially marinated fish [10].

Comment 5Lines 128-132: The sentence should probably be reworded because P<0.05 does not apply in the case of the Bligh–Dyer method.

Response 5:  We thank the Reviewer for this accurate observation. The sentence has been  revised to clearly indicate that the difference in lipid content was statistically significant when determined using  the Soxhlet method (p < 0.05) but not significant when using  the Bligh–Dyer method (p > 0.05).

Comment 6Lines 203-204: The sentence should be reworded because the reduction in FAA is 6% according to Table 2/line 2.

Response 6:  We thank the Reviewer for pointing out this inaccuracy. The text has been revised to reflect the actual 6.0% reduction in free amino acids consistent with the data presented in Table 2.

Comment 7Lines 189-191: "heme" instead of "heam"

Response 7:  We thank the Reviewer for the correction. The term has been changed to “heme”.

Comment 8General comment – question: The authors believe that the reduction of natural microflora through the application of preservatives may have had an effect on reduced proteolysis. In other words, the extracellular enzymes of the bacterial microflora have no effect on the final physicochemical characteristics of the product. In other words, the extracellular enzymes of the bacterial microflora have no effect on the final physicochemical characteristics of the product. I am not asking for further experiments, but if there is relevant literature, it would be good to add it.

Response 8:  We thank the Reviewer for this insightful question. We agree that microbial enzymes can, in principle, contribute to proteolysis in fish products. However, in the case of marinated herring, the combination of low pH, high salt concentration, and reduced water activity effectively  suppresses bacterial growth and extracellular enzyme activity. Previous  studies on salted and marinated herring with and without microbial inhibitors (e.g., sodium azide) have demonstrated that muscle proteolysis is mainly driven by endogenous cathepsins and aminopeptidases rather than by bacterial enzymes. Additionally, in our study the bacterial count in marinades without preservatives was very low (2 log CFU/g), indicating that the contribution of bacterial proteases was likely marginal.

Reviewer 3 Report

Comments and Suggestions for Authors

The manuscript addresses an important and underexplored area of seafood technology: the broader biochemical and sensory consequences of sodium benzoate and potassium sorbate in industrial-scale marinated herring. The study is timely given increasing regulatory pressure on Listeria monocytogenes control in ready-to-eat seafood. The experimental design is clear, with industrial-scale relevance, and multiple analytical dimensions (physicochemical, enzymatic, lipid oxidation, microbiology, sensory) were integrated. The major strength lies in linking preservative use not only to microbial stabilization but also to unintended drawbacks such as suppression of proteolysis, reduction of antioxidant peptides, and acceleration of lipid oxidation. However, some aspects require refinement.

  1. While the study demonstrates new findings on preservative-induced inhibition of proteolysis and its link to lipid oxidation, the introduction relies heavily on older literature (1960s–1980s) without clearly positioning the novelty of this work in the context of recent advances in seafood preservation. A sharper distinction between past work and the present contribution is needed.
  2. The marination period was limited to 7 days, which does not fully reflect the longer storage conditions (weeks to months) under which preservative effects may accumulate. The authors should justify why 7 days was chosen and discuss its implications for generalizing results.
  3. Several key sensory and texture outcomes are reported as non-significant but interpreted qualitatively (e.g., chemical odor, lower juiciness). With only 7 panelists, the sensory panel is underpowered. Expanding or at least acknowledging this limitation is necessary.
  4. The claim that preservatives accelerate lipid oxidation by reducing antioxidant peptides is plausible but speculative. Direct assays of antioxidant capacity (e.g., DPPH, FRAP) or peptide profiling would strengthen this argument. At minimum, limitations should be acknowledged.
  5. While preservatives clearly suppressed microbial counts, the ecological dynamics (e.g., replacement of suppressed groups by others) were not investigated. The discussion should integrate more on microbial ecology and not only total counts.
  6. Some tables are dense and difficult to interpret (e.g., proximate composition, enzyme activities). Condensing results into clearer figures or emphasizing statistically significant differences would aid readability.
  7. The conclusion recommends “cautious, minimal-use” of preservatives but does not balance this against safety benefits. The trade-off between microbial safety and nutritional quality should be discussed more critically.
  8. Typographical inconsistencies occur (e.g., “m3arinated” on page 3). Careful proofreading is required.
  9. Units should be standardized (e.g., mg/100 g, g/100 g). At times percentages and absolute values are mixed confusingly.
  10. Discussion occasionally repeats results without deeper interpretation; shortening redundancy will improve flow.
  11. The conclusion overstates findings; it should be more cautious in attributing causality between preservative use and lipid oxidation.

Author Response

Dear Reviewer 3, 

We would like to express our sincere gratitude for your thorough review of our manuscript and for the insightful comments and constructive suggestions you have provided. Your feedback has been extremely valuable in helping us improve the overall quality, clarity, and rigor of the paper. 

In this revised version of the manuscript, we have carefully considered each of your remarks. Wherever possible, we have incorporated the recommended changes, and we believe that these revisions have substantially strengthened the paper. For the sake of transparency, our responses are presented point by point, corresponding to each of the reviewers’ comments. In cases where we felt it appropriate to maintain our original approach, we have provided detailed justifications. 

We hope that the revisions adequately address your concerns and that the improved manuscript meets the high standards expected by the journal. We are truly grateful for the opportunity to revise our work and for your valuable contribution to enhancing its quality. 

Comment 1:  The manuscript addresses an important and underexplored area of seafood technology: the broader biochemical and sensory consequences of sodium benzoate and potassium sorbate in industrial-scale marinated herring. The study is timely given increasing regulatory pressure on Listeria monocytogenes control in ready-to-eat seafood. The experimental design is clear, with industrial-scale relevance, and multiple analytical dimensions (physicochemical, enzymatic, lipid oxidation, microbiology, sensory) were integrated. The major strength lies in linking preservative use not only to microbial stabilization but also to unintended drawbacks such as suppression of proteolysis, reduction of antioxidant peptides, and acceleration of lipid oxidation.

Response 1:  We sincerely thank the Reviewer for the positive and encouraging evaluation. We greatly appreciate the recognition of our study’s industrial relevance, comprehensive analytical approach, and the significance of linking preservative use to both microbiological stabilization and potential biochemical drawbacks.

However, some aspects require refinement.

Comment 2:  While the study demonstrates new findings on preservative-induced inhibition of proteolysis and its link to lipid oxidation, the introduction relies heavily on older literature (1960s–1980s) without clearly positioning the novelty of this work in the context of recent advances in seafood preservation. A sharper distinction between past work and the present contribution is needed.

Response 2:  We thank the Reviewer for this valuable observation. We agree that the Introduction could more clearly highlight the novelty of the present study in the context of recent advances in seafood preservation. In the revised version, we have expanded the introductory section to emphasize current industry trends toward the minimal use of chemical preservatives and the need to understand their broader biochemical consequences. The added sentences clarify that this study provides new insights by linking preservative-induced inhibition of proteolysis to lipid oxidation processes in industrially produced marinated herring, a relationship not previously addressed in the literature.

Comment 3:  The marination period was limited to 7 days, which does not fully reflect the longer storage conditions (weeks to months) under which preservative effects may accumulate. The authors should justify why 7 days was chosen and discuss its implications for generalizing results.

Response 3:  We thank the Reviewer for this valuable remark. The 7-day marination period was selected to reflect the typical industrial production step, after which herring fillets are packed in cover brines or sauces for long-term storage. Our aim was to evaluate whether the preservative effects are already evident during the primary marination phase, when the major biochemical transformations, including protease activation and the onset of lipid oxidation, occur most intensely. Extending the study beyond this point would introduce additional variables (e.g., the type of cover brine, oxygen availability, packaging material, or cross-contamination from personnel and/or vegetables in cover brine) that could obscure the direct effect of preservatives. We have clarified this rationale and added a brief comment in the Conclusion section regarding the potential accumulation of preservative effects during subsequent storage.

Comment 4:  Several key sensory and texture outcomes are reported as non-significant but interpreted qualitatively (e.g., chemical odor, lower juiciness). With only 7 panelists, the sensory panel is underpowered. Expanding or at least acknowledging this limitation is necessary.

Response 4:  We thank the Reviewer for this important observation. We agree that the sensory panel size (n = 7) limits the statistical power to detect subtle differences between treatments. However, the assessors were highly experienced professionals with over 10 to 15 years of practice in evaluating marinated fish products, ensuring consistent and reliable qualitative assessments. The perceived “chemical” note was consistently identified by all panelists, even though the overall sensory scores did not differ significantly. To address this, we have added a statement acknowledging the limitation of panel size and its implications for interpretation has been added to the Discussion section (Section 2.4. Colour, texture profile, and sensory evaluation of marinated herring fillets).

Comment 5:  The claim that preservatives accelerate lipid oxidation by reducing antioxidant peptides is plausible but speculative. Direct assays of antioxidant capacity (e.g., DPPH, FRAP) or peptide profiling would strengthen this argument. At minimum, limitations should be acknowledged.

Response 5:  We appreciate this insightful point. We agree that our mechanistic claim is based on inference as we did not perform direct antioxidant assays or peptide profiling. However, prior work on herring systems has demonstrated that protein and peptide fractions originating from marinating brines and muscle exhibit radical-scavenging and iron-chelating activities and can effectively retard lipid oxidation when used as coatings or additives, under acidic and saline conditions comparable to marinades. These findings support the plausibility that the reduction of peptide pools - observed here in preservative-treated samples - may have diminished endogenous antioxidant protection, thereby accelerating lipid oxidation. We have added a short paragraph to the Discussion acknowledging this limitation and citing the supporting evidence , and we note that future studies will combine peptide profiling with direct antioxidant capacity assays.

Comment 6:  While preservatives clearly suppressed microbial counts, the ecological dynamics (e.g., replacement of suppressed groups by others) were not investigated. The discussion should integrate more on microbial ecology and not only total counts.

Response 6:  We thank the Reviewer for this valuable comment. We agree that preservatives may alter not only the total microbial load but also the ecological balance of microbial groups, as previously reported by Bykowski (1983). However, the aim of the present study was not to perform a detailed microbiological characterization but to verify the technological effects of preservatives through their impact on enzymatic and chemical transformations during industrial marination. The microbiological analyses were therefore limited to total counts of psychrotrophic, mesophilic, and spoilage-indicator groups in order to confirm the general antimicrobial action of sodium benzoate and potassium sorbate.

In our data, marinated herriing with preservatives frequently fell below the plate-count limit of quantification (<10 CFU/g), which precluded any reliable inference about community shifts or ecological replacement after 7 days of  marination. While sub-detectable or viable but non-culturable (VBNC) cells cannot be excluded, the study was scoped to the industrial marination phase; potential ecological dynamics are more likely to emerge during extended storage after packaging  in cover brines, which was outside our design. We have clarified this limitation and explicitly stated that detailed microbiome profiling during storage would be a valuable direction for future work.

Comment 7:  Some tables are dense and difficult to interpret (e.g., proximate composition, enzyme activities). Condensing results into clearer figures or emphasizing statistically significant differences would aid readability.

Response 7:  We thank the Reviewer for this comment. The tables were formatted according to standard MDPI journal style, with statistically significant differences indicated by superscript letters (p < 0.05). To further improve readability, we enhanced the visual separation of data groups (e.g., colour and texture results) by expanding and bolding header rows, and clarified this in formatting the table captions. The revised layout enhances clarity while preserving data integrity and consistency with the journal’s formatting requirements .

Comment 8:  The conclusion recommends “cautious, minimal-use” of preservatives but does not balance this against safety benefits. The trade-off between microbial safety and nutritional quality should be discussed more critically.

Response 8:  We appreciate this valuable comment. The Conclusions section have been revised to better balance the discussion of safety benefits against the nutritional and sensory drawbacks of preservative use. The revised text emphasizes that preservatives, while inhibiting proteolysis and accelerating lipid oxidation, remain essential  for ensuring microbiological safety under industrial conditions. A cautious, rational approach - using the minimal effective concentration required for safety - is now explicitly recommended, along with future research directions toward alternative preservation strategies.

Comment 9:  Typographical inconsistencies occur (e.g., “m3arinated” on page 3). Careful proofreading is required.

Response 9:  We thank the Reviewer for noticing this typographical error. The indicated mistake (“m3arinated”) as well as other minor inconsistencies have been corrected. The entire manuscript has been carefully proofread for uniformity in units, symbols, and formatting to ensure overall typographical consistency.

Comment 10:  Units should be standardized (e.g., mg/100 g, g/100 g). At times percentages and absolute values are mixed confusingly.

Response 10:  We thank the Reviewer for this valuable remark. All units have been standardized throughout the manuscript. Concentrations of major components (moisture, protein, lipids, salt, acidity, and preservatives) are now consistently expressed as g/100 g, while minor nitrogen fractions are expressed  as mg/100 g to enhance numerical readability. Percentage values are used exclusively to denote relative changes between treatments. These revisions improve  consistency and clarity in data presentation.

Comment 11:  Discussion occasionally repeats results without deeper interpretation; shortening redundancy will improve flow.

Response 11:  We appreciate the Reviewer’s observation regarding occasional repetition in the Discussion. The text has been carefully reviewed to remove unnecessary redundancies and to improve flow. Some repetitions, however, were intentionally retained to ensure clarity when presenting results for different sample matrices (muscle and brine) or to emphasize specific mechanistic links (e.g., between peptide loss and lipid oxidation). These repetitions serve to maintain logical coherence and reader orientation in sections where parallel biochemical processes were discussed.

Comment 12:  The conclusion overstates findings; it should be more cautious in attributing causality between preservative use and lipid oxidation.

Response 13:  We thank the Reviewer for this important remark. We agree that the conclusion should avoid implying strict causality, as the mechanistic link between preservative use and lipid oxidation was inferred from correlated biochemical changes rather than directly demonstrated. Accordingly, the phrasing in the Conclusions has been revised to use more cautious terms such as “may” and “possibly,” indicating a plausible relationship rather than a confirmed causal effect.

Round 2

Reviewer 1 Report

Comments and Suggestions for Authors

Accept in present form